

# Identification of reference genes in blood before and after entering the plateau for SYBR green RT-qPCR studies

Jun Xiao, Xiaowei Li, Juan Liu, Xiu Fan, Huifen Lei and Cuiying Li

Department of Blood Transfusion, General Hospital of Air Force, PLA, Beijing, China

## ABSTRACT

**Background**. Tibetans have lived at high altitudes for thousands of years, and they have unique physiological traits that enable them to tolerate this hypoxic environment. However, the genetic basis of these traits is still unknown. As a sensitive and highly efficient technique, RT-qPCR is widely used in gene expression analyses to provide insight into the molecular mechanisms underlying environmental changes. However, the quantitative analysis of gene expression in blood is limited by a shortage of stable reference genes for the normalization of mRNA levels. Thus, systematic approaches were used to identify potential reference genes.

**Results**. The expression levels of eight candidate human reference genes (*GAPDH*, *ACTB*, *18S RNA*, *β2-MG*, *PPIA*, *RPL13A*, *TBP* and *SDHA*) were assessed in blood from hypoxic environments. The expression stability of these selected reference genes was evaluated using the geNorm, NormFinder and BestKeeper programs. Interestingly, *RPL13A* was identified as the ideal reference gene for normalizing target gene expression in human blood before and after exposure to high-altitude conditions.

**Conclusion**. These results indicate that different reference genes should be selected for the normalization of gene expression in blood from different environmental settings.

## INTRODUCTION

Hypoxia is a major biological feature of high-altitude regions (*Beall, 2000*). In hypoxic environments, transcription of various genes, such as endothelial PAS domain-containing protein 1 (*EPAS1*) and prolyl hydroxylase domain-containing protein 2 (*PHD2*), is initiated by hypoxia-related pathways. An increasing number of studies show that the hypoxia-inducible factor (HIF) signaling pathway plays a vital role in the adaptation to hypoxia (*Ji et al., 2012*). The human *EPAS1* gene encodes the alpha subunit of HIF-2 (HIF-2α), which acts as a key regulator of chronic hypoxia by regulating a large number of genes (*Beall et al., 2010*).

To examine the molecular mechanisms involved in these processes, quantitative gene expression analysis is indispensable. Quantitative real-time PCR (RT-qPCR) is a highly sensitive, precise and reproducible method for the detection of gene expression levels (*Bustin, 2002*; *Bustin & Nolan, 2004*; *Vandesompele et al., 2002*). However, to produce optimal results from RT-qPCR analysis, minimum requirements must be met, including

Corresponding author
Cuiying Li, licuiying2013@qq.com, 1483883255@qq.com

quality control of the mRNA and primers, PCR efficiency determination and selection of the appropriate reference genes (*Nolan, Hands & Bustin, 2006*). The obtained gene expression profile varies based on the use of different housekeeping genes as internal references genes (*Sellars et al., 2007*). Therefore, proper reference gene selection guarantees the accuracy of the analysis data obtained from RT-qPCR (*Vandesompele et al., 2002*).

Researchers have always empirically determined reference genes, such as *GAPDH* and *β-actin*, during quantitative gene expression analyses. However, recent studies have shown that housekeeping gene (HKG) expression levels vary between cell types (*Gentile et al., 2016*; *Ofinran et al., 2016*; *Wang et al., 2015*) and experimental conditions (*Tricarico et al., 2002*; *Zhang, Ding & Sandford, 2005*). Thus, a stable and suitable reference gene must be selected for the normalization of target gene expression.

In the present study, three algorithms (geNorm, NormFinder and BestKeeper) were utilized to analyze the stability of selected candidate reference genes (*glyceraldehyde-3-phosphate dehydrogenase* (*GAPDH*), *β-actin* (*ACTB*), *18S RNA*, *β2-microglobulin* (*β2-MG*), *peptidylprolyl isomerase A* (*PPIA*), *ribosomal protein L13* (*RPL13A*), *TATA-Box binding protein* (*TBP*) and s*uccinate dehydrogenase complex, subunit A* (*SDHA*)) in human blood before and after exposure to high-altitude conditions using RT-qPCR with SYBR green.

## MATERIALS AND METHODS

### Sample information

Six healthy male Han Chinese volunteers (21.3 ± 1.3 years old) who have in the plains (altitude 500 m) for at least 20 years were enrolled. Blood samples were collected when they lived in the plains and 3 days after they moved onto the plateau (altitude 4,700 m). They did not show any clinical signs of hypoxia at the time of the examination. This study was approved by the Institutional Review Board of the General Hospital of the Air Force, PLA (afgh-IRB-16-03). Each of the six volunteers provided written informed consent.

### RNA samples and cDNA synthesis

Mononuclear cells were isolated from 5 ml of peripheral blood (before and after moving to the plateau, 3,700 m) by using lymphocyte separation medium (Solarbio, Beijing, China), as previously described (*Chen et al., 2016*). Total RNA was extracted from $10^7$ mononuclear cells using TRIzol Reagent (Invitrogen, Carlsbad, CA, USA) according to the manufacturer's protocol and then quantified using a UV-2550 spectrophotometer (Shimadzu, Kyoto, Japan). cDNA was synthetized from approximately 0.5 µg of total RNA using a ReverTra Ace®qPCR RT kit with gDNA Remover (TOYOBO, Osaka, Japan).

### Candidate genes and primers for RT-qPCR

Eight candidate human reference genes, *GAPDH*, *ACTB*, *18S RNA*, *β2-MG*, *PPIA*, *RPL13A*, *TBP* and *SDHA*, were selected for evaluation based on the Minimum Information for Publication of Quantitative Real-Time PCR Experiments (MIQE) guidelines (*Bustin et al., 2009*) (Table 1). BLAST software was used to design the specific primers and to confirm the specificity of the primer sequences for the indicated gene. All primers, except for *18S RNA* and *β2-MG*, spanned one intron to exclude the contamination of genomic DNA in total RNA.

**Table 1** **Primer sequence information for RT-qPCR amplification used in this study.**

| Symbol | Gene name | Accession number | Forward primer sequence [5′–3′] | Position in cDNA | Reverse primer sequence [5′–3′] | Position in cDNA | Production size |
|---|---|---|---|---|---|---|---|
| GAPDH | Glyceraldehyde | NM_002046.5 | TCCAAAATCAAGTGGGGCGA | 4th exon | TGATGACCCTTTTGGCTCCC | 5th exon | 115 bp |
| ACTB | β-actin | NM_001101.3 | CTTCCAGCCTTCCTTCCTGG | 4th exon | CTGTGTTGGCGTACAGGTCT | 5th exon | 110 bp |
| 18s RNA | 18s RNA | M10098.1 | GGAGCCTGCGGCTTAATTTG | | CCACCCACGGAATCGAGAAA | | 100 bp |
| β2-MG | β2-microglobulin | NM_004048.2 | TGGGTTTCATCCATCCGACA | 2th exon | TCAGTGGGGGTGAATTCAGTG | 2 exon | 138 bp |
| PPIA | Peptidylprolylisomerase A | NM_021130.3 | GACTGAGTGGTTGGATGGCA | 4th exon | TCGAGTTGTCCACAGTCAGC | 5th exon | 141 bp |
| RPL13A | Ribosomal protein L13 | NM_012423.3 | AAAAGCGGATGGTGGTTCCT | 6th exon | GCTGTCACTGCCTGGTACTT | 7th exon | 118 bp |
| TBP | TATA-Box binding protein | NM_003194.4 | CAGCTTCGGAGAGTTCTGGG | 3th exon | TATATTCGGCGTTTCGGGCA | 4th exon | 117 bp |
| SDHA | Succinate dehydrogenase complex, subunit A | NM_004168.3 | AAACTCGCTCTTGGACCTGG | 10th exon | TCTTCCCCAGCGTTTGGTTT | 11th exon | 111 bp |

**Table 2** **RT-qPCR analysis for determination of the amplification efficiency.**

| Gene | Slope | $E$ (%) | $R^2$ |
|---|---|---|---|
| GAPDH | −3.162 | 107.1 | 0.999 |
| ACTB | −3.432 | 95.6 | 0.997 |
| 18s RNA | −3.422 | 96.0 | 0.998 |
| β2-MG | −3.302 | 100.8 | 0.998 |
| PPIA | −3.014 | 114.7 | 0.990 |
| RPL13A | −3.254 | 102.9 | 0.999 |
| TBP | −3.227 | 104.1 | 0.997 |
| SDHA | −3.199 | 105.4 | 0.997 |

**Notes.**
$E$, efficiency; $R^2$, correlation coefficient.

## SYBR green real-time quantitative RT-PCR

PCR was performed using a CFX-96 thermocycler PCR system (Bio-Rad, Hercules, CA, USA). In each run, 1 μl of synthetized cDNA was added to 19 μl of reaction mixture containing 8 μl of $H_2O$, 10 μl of THUNDERBIRD qPCR Mix (TOYOBO, Osaka, Japan) and 0.5 μl of forward and reverse primers (10 μM). Each sample was measured in triplicate. PCR was conducted at 95 °C for 3 min followed by 40 cycles of 95 °C for 10 s, 58 °C for 15 s and 72 °C for 15 s. The amplification was followed by melting curve analysis.

## Amplification efficiency and primer specificity of the reference genes

The amplification efficiency ($E$) of the primers was tested using a standard curve RT-qPCR of a serially diluted (1/10, 1/100, 1/1,000, 1/10,000, and 1/100,000) cDNA sample with the formula $E$ (%) $= (10^{-1/\text{slope}} - 1) \times 100$ (Ahn et al., 2008). The efficiency ($E$) and correlation coefficient ($R^2$) of each candidate reference gene were calculated to determine amplification efficiency (Table 2). An amplification efficiency of 90–110% and an $R^2$ of 0.99 were acceptable.

## Analysis of reference gene expression stability

The *geNorm* (Vandesompele et al., 2002) program was used to measure gene expression stability ($M$), and this method differs from model-based approaches by comparing genes

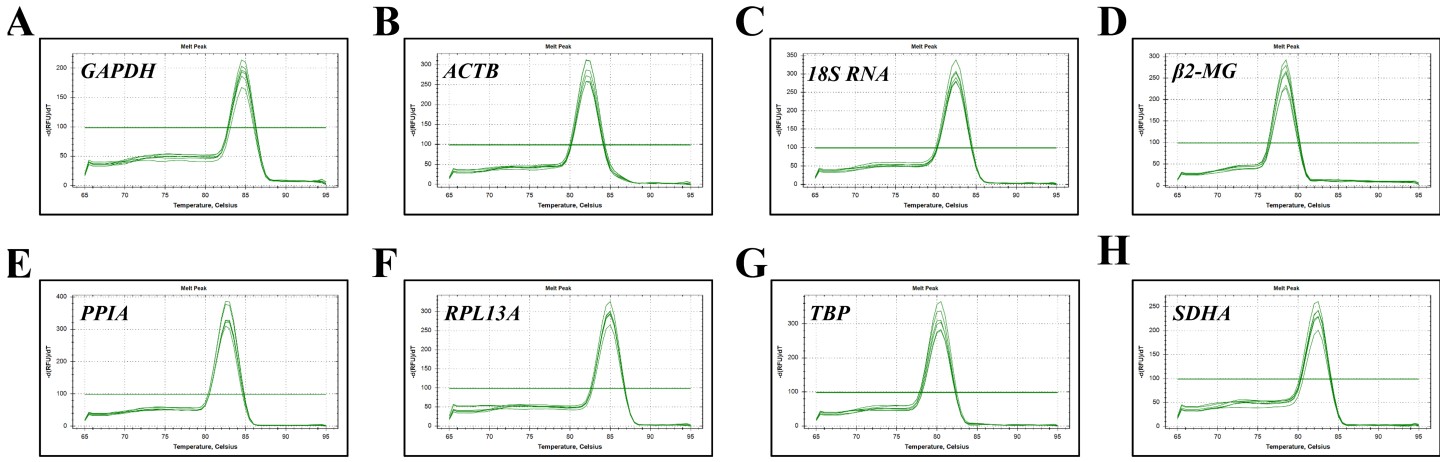

**Figure 1** **Specificity of RT-qPCR amplicons.** Dissociation curves with single peaks were generated from all amplicons and showed no dimer formation for each reference gene.

based on the similarity of their expression profiles. *geNorm* ranks the genes based on *M* values, where the gene with the most stable expression has the lowest value. *NormFinder* (*Andersen, Jensen & Orntoft, 2004*) was used to find two genes with the least intra- and inter-group expression variation. A BestKeeper index was created using the geometric mean of the *Ct* values of each candidate gene. An estimation of the reference gene stability could be obtained by analyzing the calculated variation (standard deviation and coefficient variance) (*Pfaffl et al., 2004*).

Finally, RefFinder, a comprehensive web-based tool that integrates geNorm, NormFinder and BestKeeper, was applied to determine the most stable reference gene for the final ranking (*Liu et al., 2015*).

## RESULTS

### Determining the specificity and amplification efficiency of the primers

The expression stability of eight candidate reference genes in subjects before and after migrating onto the plateau was analyzed using RT-qPCR. For each reference gene, primer specificity was demonstrated by a single peak in the melting curve analysis (Fig. 1). Amplification efficiencies were calculated as previously described (*Ahn et al., 2008*) and ranged from 95.6% to 114.7% for the eight reference genes. The correlation coefficient ($R^2$) of the standard curve for each gene was greater than 0.98 (Table 2).

### Expression levels of reference genes in the blood before and after migrating onto the plateau

To examine the stability of eight HKGs before and after migrating onto the plateau, the expression levels were evaluated by RT-qPCR, and the Shapiro–Wilk test was used to evaluate the normality of the *Ct* values (Table 3). The *Ct* values ranged from 13.40 (*ACTB*) to 21.34 (*TBP*) for the blood samples before ascending to the plateau (Table 3 and Fig. 2A) and 13.60 (*RPL13A*) to 21.78 (*TBP*) for the samples taken after ascending to the plateau

**Table 3  Descriptive statistics and normality evaluation of the reference genes *Cq* values before and after entering plateau.**

|  | Gene | Mean | SD | Min *Cq* | Max *Cq* | SW-test *p* |
|---|---|---|---|---|---|---|
| Before plateau | GAPDH | 18.41 | 0.09 | 18.31 | 18.52 | 0.248 |
|  | ACTB | 13.40 | 0.13 | 13.25 | 13.63 | 0.601 |
|  | 18s RNA | 15.84 | 0.56 | 14.79 | 16.63 | 0.989 |
|  | β2-MG | 15.86 | 0.30 | 15.46 | 16.25 | 0.326 |
|  | PPIA | 16.82 | 0.18 | 16.66 | 17.09 | 0.095 |
|  | RPL13A | 13.88 | 0.10 | 13.71 | 14.00 | 0.620 |
|  | TBP | 21.34 | 0.26 | 20.96 | 21.75 | 0.996 |
|  | SDHA | 19.79 | 0.26 | 19.41 | 20.17 | 0.963 |
| After plateau | GAPDH | 18.01 | 0.28 | 17.68 | 18.40 | 0.664 |
|  | ACTB | 14.35 | 0.43 | 13.84 | 14.89 | 0.526 |
|  | 18s RNA | 15.79 | 0.29 | 15.32 | 16.10 | 0.616 |
|  | β2-MG | 16.43 | 0.45 | 15.71 | 16.90 | 0.661 |
|  | PPIA | 16.96 | 0.30 | 16.38 | 17.26 | 0.089 |
|  | RPL13A | 13.60 | 0.15 | 13.39 | 13.77 | 0.486 |
|  | TBP | 21.78 | 0.73 | 20.50 | 22.60 | 0.530 |
|  | SDHA | 20.11 | 0.32 | 19.67 | 20.57 | 0.987 |

Notes.

SD, standard deviation; Min *Cq*, minimum *Cq* value; Max *Cq*, maximum *Cq* value; SW-test *p*, *p*-value of the Shapiro–Wilk test.

(Table 3 and Fig. 2B). *ACTB* and *RPL13A* were more abundantly expressed than the other genes before and after migrating onto the plateau (Fig. 2).

## Candidate reference gene stability: geNorm

Candidate reference gene stability was evaluated based on the $M$ values of the genes using the *geNorm* algorithm (*Vandesompele et al., 2002*). The $M$ values for *GAPDH, ACTB, 18S RNA, β2-MG, PPIA, RPL13A, TBP* and *SDHA* were lower than 1.5 in all samples. According to the analysis, *GAPDH* and *ACTB* were the most stable among all eight candidate genes on the plains (Fig. 3A), whereas *18S RNA* and *RPL13A* were the most stable genes on the plateau (Fig. 3B). Analysis of samples from both stages confirmed that *GAPDH* and *RPL13A* were the most stable genes (Fig. 3C).

Using the *geNorm* algorithm, the pairwise variation value $(V_n/V_{n+1})$ was used to calculate the optimum number of reference genes for accurate normalization and to determine whether the addition of another reference gene $(n+1)$ for normalization was recommended. A cut-off threshold $(V_n/V_{n+1} = 0.15)$ was used to determine the optimal number of reference genes required for normalization (*Vandesompele et al., 2002*). The greater the number of reference genes used for normalization, the more confidence there is in their gene expression level (*Jaramillo et al., 2017*). Two reference genes were sufficient for gene expression analysis of the blood in the plains (Fig. 3D) and plateau stages (Fig. 3E). When all samples were analyzed together, the $Vn/Vn+1$ values ranged from 0.062 to 0.110 and were all lower than the threshold value of 0.15 (Fig. 3F). Thus, only two HKGs are required for the normalization of target genes in expression analyses.
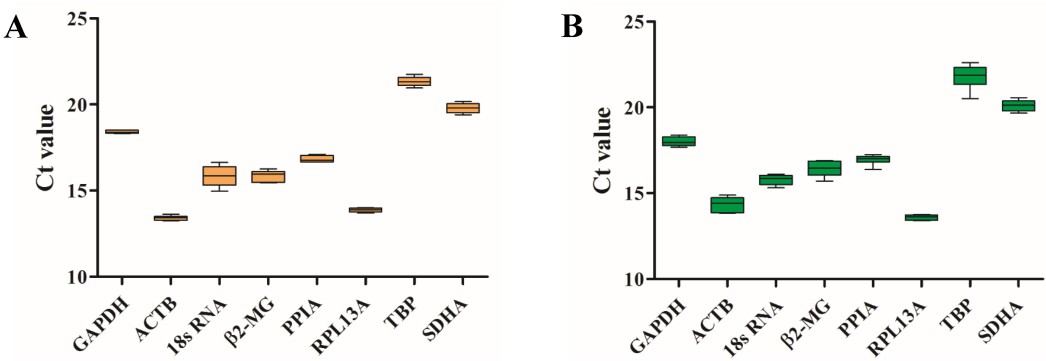

**Figure 2  Candidate reference gene *Ct* value distributions.** Boxplots of the *Ct* values from six volunteers from the plain (A) and the plateau (B) stages for each of the eight candidate reference genes.

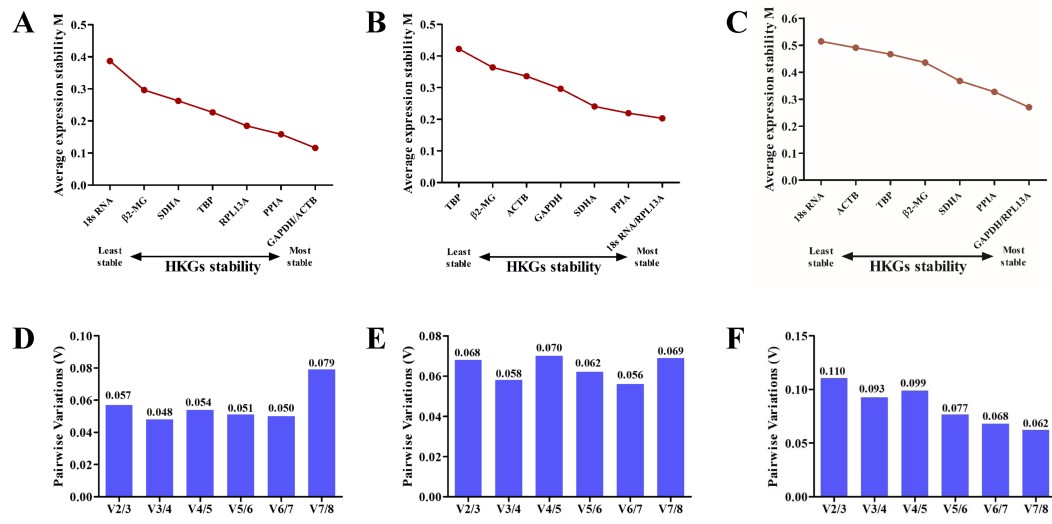

**Figure 3  The geNorm selection analysis of candidate reference genes.** The average expression stability value (*M*) was calculated by geNorm for each gene on the plain (A), plateau (B) or both stages (C). Pairwise variation (*V*) between the normalization factors (*Vn* and *Vn* + 1) was used to determine the optimal number of reference genes for normalization on the plain (D), plateau (E) or both stages (F).

## Candidate reference gene stability: NormFinder

The *NormFinder* algorithm ranks the HKGs according to the inter- and intra-group variations in expression (*Ahn et al., 2008*). The results indicated that *GAPDH*, *PL13A*, *ACTB* and *PPIA* in the plains group (Table 4) and *PPIA*, *SDHA*, *ACTB* and *RPL13A* in the plateau group (Table 4) were the most stable reference genes. *PPIA*, *SDHA*, *TBP* and *RPL13A* were the four most stable reference genes in both groups (Table 4).

## Candidate reference gene stability: BestKeeper

The *BestKeeper* algorithm (*Pfaffl et al., 2004*) uses the coefficient variance (*CV*) and standard deviation (SD) of candidate gene expression to determine the optimal HKGs (Table 5). In the *BestKeeper* program, HKGs with lower SD and *CV* values are considered as optimal reference genes. In both stages, *RPL13A* expression had the lowest SD (0.15) and
**Table 4  Calculation of candidate reference genes expression stability by the *NormFinder*.**

| Ranking order | Gene | Stability value (Whole stages) | Gene | Stability value (Plain) | Gene | Stability value (plateau) |
|---|---|---|---|---|---|---|
| 1 | PPIA | 0.080 | GAPDH | 0.032 | PPIA | 0.076 |
| 2 | SDHA | 0.136 | RPL13A | 0.057 | SDHA | 0.166 |
| 3 | TBP | 0.205 | ACTB | 0.110 | ACTB | 0.176 |
| 4 | RPL13A | 0.227 | PPIA | 0.121 | RPL13A | 0.182 |
| 5 | 18s RNA | 0.229 | TBP | 0.167 | 18s RNA | 0.194 |
| 6 | β2-MG | 0.237 | SDHA | 0.210 | GAPDH | 0.237 |
| 7 | GAPDH | 0.265 | β2-MG | 0.264 | β2-MG | 0.244 |
| 8 | ACTB | 0.316 | 18s RNA | 0.434 | TBP | 0.369 |

**Table 5  Results from BestKeeper analysis.**

| Gene | | Whole stage | Plain | Plateau |
|---|---|---|---|---|
| GAPDH | std dev [±CP] | 0.24 | 0.08 | 0.23 |
| | CV [% CP] | 1.33 | 0.41 | 1.25 |
| ACTB | std dev [±CP] | 0.48 | 0.11 | 0.35 |
| | CV [% CP] | 3.45 | 0.82 | 2.45 |
| 18s RNA | std dev [±CP] | 0.33 | 0.44 | 0.23 |
| | CV [% CP] | 2.09 | 2.79 | 1.45 |
| β2-MG | std dev [±CP] | 0.37 | 0.26 | 0.33 |
| | CV [% CP] | 2.31 | 1.63 | 2.04 |
| PPIA | std dev [±CP] | 0.21 | 0.16 | 0.19 |
| | CV [% CP] | 1.24 | 0.93 | 1.15 |
| RPL13A | std dev [±CP] | 0.15 | 0.09 | 0.12 |
| | CV [% CP] | 1.10 | 0.62 | 0.89 |
| TBP | std dev [±CP] | 0.43 | 0.22 | 0.53 |
| | CV [% CP] | 1.98 | 1.01 | 2.42 |
| SDHA | std dev [±CP] | 0.27 | 0.23 | 0.25 |
| | CV [% CP] | 1.34 | 1.15 | 1.22 |

the lowest *CV* (1.10). Therefore, *RPL13A* was proposed as the ideal HKG for the analysis of gene expression during the plains and plateau stages.

## Candidate reference gene stability: RefFinder

Based on the geNorm, NormFinder and BestKeeper results, RefFinder (http://leonxie.esy.es/RefFinder/) was used to calculate a comprehensive expression stability ranking. As shown in Table 6, *GAPDH* (plains) and *PPIA* (plateau) were the most stable HKGs before and after entering the plateau, respectively. Across both stages, *PPIA* and *RPL13A* were the most stable reference genes for the normalization of target gene expression levels.

## DISCUSSION

Understanding the mechanisms of high-altitude hypoxic adaptation is a major focus of high-altitude medical research. Using RT-qPCR to rapidly and accurately analyze

**Table 6   Stabilities of HKGs ranked by RefFinder.**

| Ranking order | Whole stages | Plain | Plateau |
|---|---|---|---|
| 1 | PPIA | GAPDH | PPIA |
| 2 | RPL13A | RPL13A | RPL13A |
| 3 | SDHA | ACTB | 18sRNA |
| 4 | GAPDH | PPIA | SDHA |
| 5 | β2-MG | TBP | GAPDH |
| 6 | TBP | SDHA | ACTB |
| 7 | 18sRNA | β2-MG | β2-MG |
| 8 | ACTB | 18sRNA | TBP |

gene expression is a common strategy for understanding the mechanisms of this process (*Valasek & Repa, 2005*). Since the expression levels of reference genes in endothelial cells (*Bakhashab et al., 2014*), epithelial cells (*Liu et al., 2016*) and cancer cells (*Fjeldbo et al., 2016*; *Lima et al., 2016*) can vary under hypoxic conditions, gene expression was analyzed in blood from subjects at various altitudes to determine which reference genes should be used under particular conditions. Most expression studies of blood under hypoxic conditions have used a single traditional reference gene, such as *GAPDH, ACTB* and *18S RNA* (*Polotsky et al., 2015*; *Srikanth et al., 2015*), without evaluating the expression stability of these reference genes. Therefore, it is necessary to estimate the stability of reference genes at various altitudes.

In the present study, eight different reference genes were selected to be assessed and validated for stability at different altitudes using the geNorm, NormFinder, BestKeeper and RefFinder programs. The study identified two candidate genes (*PPIA* and *RPL13A*) that are stably expressed under hypoxic stress and can be used as reference genes for relative gene quantification and normalization before and after entering the plateau region.

In this study, three widely used algorithms (geNorm, NormFinder and BestKeeper) were applied to calculate the stability of the selected reference gene expression levels. The geNorm algorithm uses the principle that the expression ratio of two ideal reference genes is identical in all tested samples (*Vandesompele et al., 2002*). According to the average pairwise variation of one reference gene with all other candidate genes, a lower $M$-value indicates greater stability of the candidate gene. NormFinder, which is based on the stability value of the internal control genes, can select the minimally fluctuating genes as the most stable genes, but it can only select one suitable reference gene for normalization. The ranking results varied across the different algorithms. The comprehensive RefFinder ranking indicated that *GAPDH* and *PPIA* were the most stable genes in the plains and plateau groups, respectively, and *PPIA* was the most stable gene in both stages.

Previous studies have reported that *β2-MG* levels do not vary with oxygen concentration (*Petousi et al., 2014*). Studies in bladder cancer cells under hypoxia showed that *β2-MG* and *Hypoxanthine phosphoribosyltransferase-1* (*HPRT*) were the most suitable reference genes for normalizing gene expression (*Lima et al., 2016*). In human retinal endothelial cells, *TBP* and *pumilio RNA binding family member 1* (*PUM1*) were the most stable reference genes under hypoxic conditions (*Xie et al., 2016*). However, the present study showed that the

stress-specific candidate genes *β2-MG* and *TBP* were not suitable for normalizing target gene expression in blood under normoxic and hypoxic conditions.

Under normoxic conditions, *GAPDH* was the most stable gene in the blood, whereas under hypoxic conditions, *PPIA* was the most stable candidate reference gene. *RPL13A* was ranked as the second most stable reference gene in blood both under normoxic and hypoxic conditions. *ACTB* was observed to be the most stable candidate gene in plain blood using the geNorm algorithm (Fig. 3B), but it was the least stable (Fig. 3A) in the combined analysis of tested samples. In the plateau stage but not in the plains stage, *18S RNA* was one of the most stable genes. The differences in the reference gene rankings could be associated with the algorithms used by each program.

Our study has some limitations. The identification of stable candidate genes for target gene expression analysis in human blood between low- and high-altitude conditions was a major challenge due to the difficulty involved in sample collection. This difficulty may account for the limited number of volunteers enrolled in the present study and the limited number of gene expression studies of blood in the plateau environment. Thus, one of the limitations of this study was that we could not collect enough blood samples to strengthen the reliability of the present study. In addition, analyses of the stability of reference gene expression should be verified at a cellular level in a hypoxic chamber. In the present study, however, the stability of candidate reference genes was reliably evaluated in blood under normoxic and hypoxic stress conditions using algorithms. Previous studies on target gene expression analyses of blood under hypoxic conditions used *18S RNA* (*Mishra et al., 2013*) and *β2-MG* (*Petousi et al., 2014*) as reference genes for normalization. The present study clearly showed that both *PPIA* and *RPL13A* are stable and suitable reference genes, but the amplification efficiency of *PPIA* was more than 1.05 (Table 2). Thus, *RPL13A* is the most suitable and stable reference gene for the normalization of target gene expression in blood from the plains and plateau environments.

In conclusion, the present study determined that *GAPDH* and *RPL13A* in blood from the plains region and *PPIA* and *RPL13A* in blood from the plateau region were the most stable reference genes. Among the identified stably expressed reference genes in both the plains and plateau environments, RPL13A was shown to be most stable in blood from both the normoxic and hypoxic conditions. Additional studies should be conducted on the cellular level to verify the stability of the same reference genes.

## CONCLUSIONS

In this study, the expression levels of eight candidate human reference genes (*GAPDH*, *ACTB*, *18S RNA*, *β2-MG*, *PPIA*, *RPL13A*, *TBP* and *SDHA*) were assessed in blood from hypoxic environments. We determined, for the first time, that *RPL13A* was the most reliable reference gene for the normalization of target gene expression in human blood from low- and high-altitude environments. However, to obtain reliable data, the use of more than one reference gene is strongly recommended.

## ACKNOWLEDGEMENTS

The authors thank Fengyan Fan for excellent technical assistance and we also thank the members of the Department of Gene Detection in our hospital for helpful discussion.

### Funding

This work was supported by the Application Fundamental Research project foundation of China (No. AWS13J004). The funders had no role in study design, data collection and analysis, decision to publish, or preparation of the manuscript.

### Grant Disclosures

The following grant information was disclosed by the authors:
Application Fundamental Research project foundation of China: AWS13J004.

### Competing Interests

The authors declare there are no competing interests.

### Author Contributions

- Jun Xiao conceived and designed the experiments, performed the experiments, analyzed the data, contributed reagents/materials/analysis tools, wrote the paper, prepared figures and/or tables, reviewed drafts of the paper.
- Xiaowei Li conceived and designed the experiments, performed the experiments, analyzed the data, contributed reagents/materials/analysis tools, prepared figures and/or tables, reviewed drafts of the paper.
- Juan Liu performed the experiments, contributed reagents/materials/analysis tools, reviewed drafts of the paper.
- Xiu Fan and Huifen Lei performed the experiments, reviewed drafts of the paper.
- Cuiying Li performed the experiments, analyzed the data, reviewed drafts of the paper.

### Human Ethics

The following information was supplied relating to ethical approvals (i.e., approving body and any reference numbers):

The General Hospital of the Air Force, PLA granted ethical approval to carry out the study within its facilities.

### Data Availability

The raw data has been supplied as a Supplemental File.

### Supplemental Information

Supplemental information for this article can be found online at http://dx.doi.org/10.7717/peerj.3726#supplemental-information.

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
