# Peer review of "Identification of reference genes in blood before and after entering the plateau for SYBR green RT-qPCR studies"

_PeerJ, doi:10.7717/peerj.3726_

## Round 0.1 · original submission · Major Revisions

Both independent reviewers have noted that more information on the experimental design needs to be added, especially regarding your healthy subjects (demographic data, time spent on the plateau before blood sampling). They also raised several other points, as indicated below. Please address their comments carefully before resubmitting.

I alos would like to add some points myself:

- The description of PCR methods also needs more detail (e.g., stating that 0.5 microliters of primer were used is meaningless unless the concentration is also indicated; PCR cycling conditions should be indicated, etc).

- Lines 69-71: The article by Meira-Strejevitch has been retracted and should not be cited, since the data may not be valid. The article by Jaramillo (2017) refers to a non-human organism - please consider choosing more appropriate references here, possibly including a recent review article that summarizes some of the existing studies on qPCR reference genes

- Importantly, please confirm that appropriate ethics approval was obtained for this study involving human subjects.

- lines 138-140: "Cq values" -is this a typo, since in Figure 2, you use the more common term "Ct values"

·

Basic reporting

─ With respect to the volunteers the clinical and experimental information should be expanded and some basic characteristics should be provided, e.g. age, sex, time spend in the plain before moving and time spent on the plateau before blood was drawn. Did any of the six volunteers they show clinical signs of hypoxia at the time of examination?
─ There are some minor typos that should be corrected, e.g. line 86 “qRCR RT kit”, line 129 “quantitative quantification method”.

Experimental design

─ The limited number of volunteers the experiments are performed on should be discussed critically.

Validity of the findings

─ Please report the number of replicates the measurements of the different housekeeping genes was performed on.
─ What do the authors think is the reason that these housekeepers reliability is different depending on the evaluated altitude?

Additional comments

In the presented study, the authors assessed the stability of eight different human housekeeping genes in standard SYBR Green RT-qPCR under normoxic and hypoxic conditions. Different algorithms (geNorm, NormFinder, BestKeeper and RefFinder) to evaluate the stability were applied. In their study, they identified RPL13A as most reliable housekeeping gene for evaluation of gene expression in normoxic and hypoxic conditions.

Reviewer 2 ·

Basic reporting

English proficiency requires improvement throughout the paper.
Examples:
-line 44: In the abstract: "Interestingly, RPL13A was selected as the
ideal reference gene to normalize the target gene expression in human blood before and
after moving to the plateau." I suggest changing the selected to identified, or found ...
- line 52: What do you mean by "Hypoxia is a major geographic feature of high-altitude regions", the word geographic does not fit here, may replace with biological?
- line 53: the following sentence is not clear: "In hypoxic environment, specific genetic programs and molecular mechanisms initiate various genetic events."
- line 54: "hypoxia-inducible factor (HIF) oxygen-signaling pathway" do you mean alpha not oxygen?
- line 154: what is (Vn / Vn + 1 = 0.15)??? please explain.
-line 172: "is based on the coefficient variance (CV) and standard deviation (SD)" of what? please complete the sentence.
- line 203: what do you mean by: "at both stages of entering the plateau"?
-Figure 3: A, B , and C, why it is connecting line, these are separate values, are they related to each other?
-table 5, 6, what do you mean by whole stage and whole stages, is it plain and plateau combined?

Experimental design

suggestions to authors:

please provide more details related to the study design, how many samples were collected from each individual and when, how many days after they were moved to the plateau.

Validity of the findings

Overall, the idea of the study is interesting and might be useful for future researchers in the field of hypoxia and the selection of appropriate housekeeping genes.
Yet, the authors do not explain the variability in their findings related to the different ranking scores among the different validation programs they used. This limitation of the study should be addressed in their discussion. Also further discussion should focus on the limitation and reliability of each of the used programs (geNorm, NormFinder and BestKeeper).

Would it be feasible for researchers to validate their finding in an experiment where cells are placed in hypoxic chamber and the same reference genes are assessed?

---

## Round 0.2 · Major Revisions

Dear authors,

Thanks for submitting a revised version of your manuscript.

Unfortunately, the reviewers still had substantial and valid comments.

Also, I agree with both reviewers that the manuscript requires language editing. Please try to address all the reviewers comments and also have the manuscript revised by a native English speaker.

·

Basic reporting

The English - including the newly added parts - would still benefit from improvements.

Experimental design

no comment

Validity of the findings

no comment

Reviewer 2 ·

Basic reporting

The manuscript is still unclear, professional English editing is critical and required for this manuscript before being accepted.

examples:
Line 78: what do you mean by “Gene sequences were deposited in the NCBI database under GenBank accession numbers indicated in Table 1”

Line 96: “correlation coefficient (R2 ) of each candidate reference gene were calculated for (Table 2)” calculated for what???

Line 100: “The geNorm (Vandesompele et al. 2002) program is a measurement of gene expression stability 101 (M)” the program is not a measurement.

Line 113: add “in subjects” after the word reference genes

Line 138: need to explain in the text what the formula means, what V is.

Line 181: “However, there are few studies on the evaluation of optimal reference gene(s) 182 between low- and high-altitude conditions” this sentence is redundant.

Experimental design

details are still lacking:

RNA samples and cDNA synthesis: this section still lacks a lot of details, such as how samples were processed, were the mononuclear cells isolated? What method as used? Or did they pelleted the whole blood? How did they extract the RNA

There is no positive control to confirm that hypoxia has occurred in the blood of the subjects after 3 days of being in the plateau. A measurement of hypoxia related gene is needed, and/or validation of these data in experimental hypoxic conditions

Validity of the findings

fig. 1: why there is no band for TBP and very faint band for SDHA? and there was no mention in the text regarding this

Table 5: what is CP?

Discussion: it does not address the limitation of the study, the tools that are used, and the significance of their finding. It is more or less repetition of the results.

Additional comments

Since the experiments related to validating these findings in hypoxic chamber is underway, these data should be feasible to add.

---

## Round 0.3 · accepted · Accept

I feel that you have now addressed the reviewers' comments adequately, and that your paper is now suitable for publication